# Extracellular vesicles derived from DFO-preconditioned canine AT-MSCs reprogram macrophages into M2 phase

Su-Min Park[1], Ju-Hyun An[1], Jeong-Hwa Lee[1], Kyung-Bo Kim[1], Hyung-Kyu Chae[1], Ye-In Oh[1], Woo-Jin Song[2]*, Hwa-Young Youn[1]*

1 Department of Clinical Veterinary Science, Laboratory of Veterinary Internal Medicine, College of Veterinary Medicine, Seoul National University, Seoul, Republic of Korea, 2 Laboratory of Veterinary Internal Medicine, College of Veterinary Medicine, Jeju National University, Jeju, Republic of Korea

* ssong@jejunu.ac.kr (WJS); hyyoun@snu.ac.kr (HYY)

**Data Availability Statement:** All relevant data are within the manuscript and its Supporting Information files.

## Abstract

### Background

Mesenchymal stem/stromal cells (MSCs) are effective therapeutic agents that ameliorate inflammation through paracrine effect; in this regard, extracellular vesicles (EVs) have been frequently studied. To improve the secretion of anti-inflammatory factors from MSCs, pre-conditioning with hypoxia or hypoxia-mimetic agents has been attempted and the molecular changes in preconditioned MSC-derived EVs explored. In this study, we aimed to investigate the increase of hypoxia-inducible factor 1-alpha (HIF-1α)/cyclooxygenase-2 (COX-2) in deferoxamine (DFO)-preconditioned canine MSC (MSC$^{DFO}$) and whether these molecular changes were reflected on EVs. Furthermore, we focused on MSC$^{DFO}$ derived EVs (EV$^{DFO}$) could affect macrophage polarization via the transfer function of EVs.

### Results

In MSC$^{DFO}$, accumulation of HIF-1α were increased and production of COX-2 were activated. Also, Inside of EV$^{DFO}$ were enriched with COX-2 protein. To evaluate the transferring effect of EVs to macrophage, the canine macrophage cell line, DH82, was treated with EVs after lipopolysaccharide (LPS) stimulation. Polarization changes of DH82 were evaluated with quantitative real-time PCR and immunofluorescence analyses. When LPS-induced DH82 was treated with EV$^{DFO}$, phosphorylation of signal transducer and transcription3 (p-STAT3), which is one of key factor of inducing M2 phase, expression was increased in DH82. Furthermore, treated with EV$^{DFO}$ in LPS-induced DH82, the expression of M1 markers were reduced, otherwise, M2 surface markers were enhanced. Comparing with EV$^{DFO}$ and EV$^{non}$.

### Conclusion

DFO preconditioning in MSCs activated the HIF-1α/COX-2 signaling pathway; Transferring COX-2 through EV$^{DFO}$ could effectively reprogram macrophage into M2 phase by promoting the phosphorylation of STAT3.

**Funding:** This research was supported by Basic Science Research Program through the National Research Foundation of Korea (NRF) funded by the Ministry of Education in the form of funds granted to WJS [2019R1A6A1A10072987].

**Competing interests:** The authors have declared that no competing interests exist.

## Introduction

Among the secretomes of mesenchymal stem/stromal cells (MSCs), extracellular vesicles (EVs) have been studied frequently for playing an important role in transmitting signals across cells [1]. EVs are small membrane vesicles that are 40–100 nm in diameter. They are released from cell membrane after fusion with multivesicular endosome and cell membrane, and contain proteins, metabolites, and nucleic acids [2]. Studies have shown the cells to be modified via transmission of mRNA, RNA, and protein via EVs [3,4].

Further studies are being conducted on how to increase the productivity and efficacy of EVs [5]. Some studies have been conducted using hypoxia-preconditioned methods to improve anti-inflammatory effect of MSC-derived EVs [6]. Several studies have shown deferoxamine (DFO), a hypoxia mimetic agent, to be usable in hypoxia preconditioning [7] and improve angiogenesis effect of MSC-derived EVs [8]. DFO inhibits hydroxylation of hypoxia-inducible factor-1 alpha (HIF-1α) and chelates the iron necessary for prolyl-4 hydroxylase, owing to which, HIF-1α is accumulated in the cell nucleus similar to that in hypoxic condition [9]. Although some previous papers had reported the modification of stem cells in hypoxic culture [10,11], there have not been many studies reporting the changes in MSC-derived EVs when treated with DFO.

Hypoxic stimulation induces HIF-1α accumulation in the nucleus, and control various signal pathways, such as inflammation, energy deprivation, or proliferation [12]. We focused on the activation of cyclooxygenase-2 (COX-2)/prostaglandin E2 (PGE 2) synthase axis [13] by HIF-1α, since the former has anti-inflammatory function in stem cells [14]. Moreover, COX-2 and signal transducer and activator of transcription 3 (STAT3) are linked to macrophages polarization [15], which is the main signal molecule in M2 polarization [16,17].

Therefore, in this study, we aimed to investigate a hypoxic culture method using DFO in order to increase the anti-inflammatory efficacy of MSC-derived EVs. Moreover, we investigated the association of HIF-1α/COX-2 anti-inflammatory pathway in DFO preconditioned canine adipose tissue derived MSC (cAT-MSC). Further, we revealed the molecular changes of MSCs to be reflected on the derived EVs, the latter then transferring the molecules to macrophages and reprogramming them into M2 phase.

## Materials and methods

### Cell preparation and culture

Canine adipose tissues (cAT) were obtained using a protocol approved by the Institutional Animal Care and Use Committee of Seoul National University (SNU; protocol no. SNU-180621-27). Briefly, Canine adipose tissue was obtained from a healthy dog < 1 year old during routine spay surgery. The tissue was washed three times with phosphate-buffered saline (PBS; PAN Biotech, Aidenbach, Germany) which was contained 100 U/ml penicillin and 100 g/ml streptomycin. Then the tissues were cut into small pieces and digested for 1 h at 37°C by collagenase type IA (1 mg/ml; Sigma-Aldrich, St. Louis, MO, USA). After 1 hour, the collagenase was inhibited with Dulbecco's Modified Eagle's Medium (DMEM; PAN Biotech) with 10% fetal bovine serum (FBS; PAN Biotech). To remove debris, the cell pellet was obtained after centrifugation at 1200 × $g$ for 5 min and filtered through a 70-μm Falcon cell strainer (Fisher Scientific, Waltham, MA, USA). Then cells were incubated in DMEM containing 10% FBS at 37°C in a humidified atmosphere of 5% $CO_2$.

cAT-MSCs were characterized as described in Supporting information. Cells were differentiated into adipocytes, chondrocytes, and osteocytes to confirm their multilineage features. They were also characterized by detecting stem cell markers with flow cytometry. After characterization, cAT-MSCs at passages 3–4 were used for subsequent experiments.

Cells were cultured in Dulbecco's modified Eagle's medium (DMEM; PAN Biotech, Aidenbach, Germany) with 10% fetal bovine serum (FBS; PAN Biotech) and 1% penicillin-streptomycin (PS; PAN Biotech) at 37˚C in 5% $CO_2$ atmosphere. The culture medium was changed every 2–3 days, and cells were sub-cultured at 70–80% confluency. When cAT-MSCs were approximately 70% confluent, 100 μM DFO was added for 48 h in DMEM with 10% exosome-depleted FBS (Thermo Fischer Scientific, Massachusetts, USA) and 1% PS (PAN Biotech).

The canine macrophage cell line DH82 was purchased from the Korean Cell Line Bank (Seoul, Korea) and cultured in DMEM with 15% FBS at 37˚C in 5% $CO_2$ atmosphere until they reached 70–80% confluency.

## Transfection of cAT-MSCs with siRNA

When the confluency of cAT-MSC was approximately 40%, they were transfected with COX-2 siRNA or control siRNA (sc-29279 and sc-37007, respectively; Santa Cruz Biotechnology, Santa Cruz, CA, USA) for 48 h using Lipofectamine RNAiMAX (Invitrogen, Carlsbad, CA, USA), following the manufacturers' instructions. COX-2 knockdown was confirmed by qRT-PCR before further experiments.

## Isolation and characterization of EVs derived from canine adipose tissue-derived (cAT)-MSCs

cAT-MSCs were cultured for 48 h in DMEM with 10% exosome-depleted FBS (Thermo Fischer Scientific) and 1% PS (PAN Biotech). The medium from each cultured cAT-MSC sample was collected and centrifuged at 2600 × g for 20 min to remove cells and cell debris. Each supernatant was transferred to a fresh tube and appropriate volume of ExoQuick-CG (System Biosciences, USA) added. EVs were isolated according to the manufacturer's instructions.

Protein markers of isolated EVs were identified by western blotting using antibodies against CD81 (Aviva system biology, CA, USA) and CD9 (GeneTex, Irvine, CA, USA). Morphology of the EVs was characterized using transmission electron microscopy. Briefly, 10 μl of EV suspension was placed on a 300-mesh formvar/carbon-coated electron microscopy grid with the coated side facing the suspension. Distilled water was placed on the mesh for washing and a 10-μl drop of uranyl acetate was placed on the mesh for negative staining for 1 min, followed by observation under a transmission electron microscope (TEM; LIBRA 120, Carl Zeiss, Germany) at 120 kV. Size distribution of the particles was determined using a zeta-potential and particle size analyzer (ELSZ-1000ZS, Otsuka Electronics, Osaka, Japan).

## RNA extraction, cDNA synthesis, and quantitative real-time polymerase chain reaction (qRT-PCR)

Total RNA was extracted from cAT-MSCs preconditioned with DFO or from control group, and from DH82 cells using the Easy-Blue total RNA extraction kit (Intron Biotechnology, Sungnam, Korea). cDNA was synthesized using the CellScript All-in-One 5× first strand cDNA synthesis master mix (CellSafe, Seoul, Korea). Samples were analyzed using AMPI-GENE qPCR green mix Hi-ROX with SYBR Green dye (Enzo Life Sciences, Farmingdale, NY, USA) and 400 nM forward and reverse primers (Table 1) in the qRT-PCR thermal cycler (Bionics, Seoul, Korea). The expression level of each gene was normalized to that of glyceraldehyde 3-phosphate dehydrogenase (GAPDH), and relative expression calculated against the contrasting control group.

**Table 1. Sequences of PCR primers used in this study.**

| Target gene | Primer | Sequence | Size |
|---|---|---|---|
| Canine | Forward | ACT GAT GAC CAA CAA CTT GAG G | 122 |
| HIF-1α | Reverse | TTT GGA GTT TCA GAA GCA GGT A | |
| Canine | Forward | TTC CTG CGA AAT ACA ATT ATG AAA T | 149 |
| COX-2 | Reverse | GCC GTA GTT CAC ATT ATA AGT TGG T | |
| Canine | Forward | TTA ACT CTG GCA AAG TGG ATA TTG T | 85 |
| GAPDH | Reverse | GAA TCA TAC TGG AAC ATG TAC ACC A | |
| Canine | Forward | AGT TGC AAG TCT CCC ACC AG | 177 |
| IL-1b | Reverse | TAT CCG CAT CTG TTT TGC AG | |
| Canine | Forward | GGC TAC TGC TTT CCC TAC CC | 243 |
| IL-6 | Reverse | TGG AAG CAT CCA TCT TTT CC | |

## Protein extraction, cell fractionation, and western blotting

Protein was extracted from preconditioned cAT-MSC-derived exosomes and DH82 using the Pro-Prep protein extraction solution (Intron Biotechnology). Concentration of the protein samples was analyzed using the DC Protein Assay Kit (Bio-Rad, Hercules, CA, USA). Nuclear proteins were isolated using the Cell Fractionation Kit-Standard (Abcam, Cambridge, MA, USA). For western blot assays, 25 μg of proteins were loaded and separated by SDS-PAGE. Bands from SDS-PAGE were transferred to polyvinylidene difluoride membranes (EMD Millipore, Billerica, MA, USA), which were then blocked with 5% non-fat dry milk and Tris-buffered saline. Membranes were incubated with primary antibodies against HIF-1α (1:500; LifeSpan BioSciences, Seattle, WA, USA), COX-2 (1:500, Santa Cruz Biotechnology, Dallas, TX, USA), STAT3 (1:500, LifeSpan BioSciences), phosphorylated (Tyr705) STAT3 (1:500, LifeSpan BioSciences), lamin A (1:500, Santa Cruz Biotechnology) and β-actin (1:1000, Santa Cruz Biotechnology) at 4˚C overnight. The membranes were subsequently incubated with the appropriate secondary antibody for 1 h. Using an enhanced chemiluminescence detection kit (Advansta, Menlo Park, CA, USA), immunoreactive bands were detected and normalized to the housekeeping protein (β-actin).

## IF analyses

DH82 cells were cultured at $2 \times 10^5$ cells in cell-culture slide (SPL, Korea), and 200 ng/ml lipopolysaccharides (LPS; Sigma-Aldrich) were stimulated for 24 h. After LPS stimulation, DH82 cells were treated with EVs at concentrations of 50 μg/well for 48 h. The slide was fixed with 4% paraformaldehyde and blocked with a blocking buffer containing 5% bovine serum albumin and 0.3% Triton X-100 (both from Sigma-Aldrich) for 1 h. They were then incubated overnight at 4˚C with antibodies against FITC-conjugated CD206 (1:200; Santa Cruz Biotechnology) and phycoerythrin-conjugated CD11b (1:100; Abcam). After three washes, the slides were mounted in a VECTASHIELD mounting medium containing 4',6-diamidino-2-phenylindole (Vector Laboratories, Burlingame, CA, USA). The samples were observed under a EVOS FL microscope (Life Technologies, Darmstadt, Germany). Immunoreactive cells were calculated with 20 random fields per group as per the ratio of DAPI/CD206-positive cells.

## Statistical analyses

Data are shown as mean ± standard deviation. Mean values from the different groups were compared by Student's t-test and one-way analysis of variance using GraphPad Prism v.6.01

software (GraphPad Software, La Jolla, CA, USA). P value < 0.05 was considered statistically significant.

## Results

### Characterization of cAT-MSC EVs

cAT-MSCs were characterized by flow cytometry and differentiation (Fig 1A–1G). EVs were separated from stem cell culture media by Exo-quick[TM]. They were round in shape, with diameter ranging from 50 nm to 100 nm, as per electron microscopic analysis (Fig 1H). Using a particle-size analyzer, the diameter of EVs was confirmed to be approximately 100 nm (Fig 1I). To identify the surface markers of exosomes, CD81 and CD 9 were confirmed as positive while β-actin was negative in western blotting (Fig 1J).

### Elevation of HIF-1α/COX-2 expression in DFO-preconditioned MSCs

In CCK assay, the viability of cAT-MSCs was found to be decreased upon treatment with 1 mM DFO; therefore, we decided to use DFO < 1 mM in further experiments (Fig 2A). We confirmed that HIF-1α accumulated in DFO-preconditioned cAT-MSCs (cAT-MSC$^{DFO}$) at both 100 μM and 500 μM of DFO concentration (Fig 2B). Considering the previous report, which found no significant difference between 100 μM and 500 μM DFO [18], 100 μM DFO was chosen for the current treatment.

To evaluate the role of COX-2 in cAT-MSCs, siCOX-2 was transfected into the cells before DFO conditioning. RNA levels of COX-2 were significantly increased in cAT-MSC$^{DFO}$ and DFO preconditioning plus control siRNA (cAT-MSC$^{siRNA}$) groups; the expression not being significantly different between the two. In DFO-preconditioned group, with COX-2 siRNA (cAT-MSC$^{siCOX-2}$), COX-2 expression was not increased (Fig 2C). At protein level, COX-2 was increased in cAT-MSC$^{DFO}$ and cAT-MSC$^{siRNA}$, but not in cAT-MSC$^{siCOX-2}$ (Fig 2D).

### cAT-MSC-derived EVs transport COX-2 to DH82 and activate the phosphorylation of STAT3

When expression of COX-2 protein was evaluated in EVs, it was found increased in cAT-MSC$^{DFO}$-derived EV (EV$^{DFO}$) than in non-preconditioned cAT-MSC-derived EV (EV$^{non}$).

In cAT-MSC$^{siCOX-2}$-derived EV (EV$^{siCOX-2}$), COX-2 was not increased, similar to that in cAT-MSCs (Fig 3A).

DH82 cells were treated with EVs to verify their anti-inflammatory effect on macrophages. They were treated with LPS before EV treatment. In the LPS-treated group, expression of p-STAT3 was increased, whereas that of STAT3 decreased compared to that in naïve group. The expression of both STAT3 and p-STAT3 increased in EV-treated groups. In the groups treated with EV$^{DFO}$, the expression of p-STAT3 was significantly increased than in non-preconditioned group. However, in the group treated with EV$^{siCOX-2}$, p-STAT3 was not increased compared to that in the group treated with EV$^{non}$ (Fig 3B).

### Change of polarization of DH82 when treated with preconditioned EVs

When DH82 cells were treated with LPS, the markers of M1 proinflammatory phase, namely IL-1b and IL-6, were significantly increased. However, expression of both cytokines was decreased in the groups with EV treatment. Expression of pro-inflammatory cytokines was decreased in the groups treated with EV$^{DFO}$, than in the groups treated with EV$^{non}$ group. The group with EV$^{siCOX-2}$ showed no significant difference from that with EV$^{non}$ (Fig 4A and 4B).

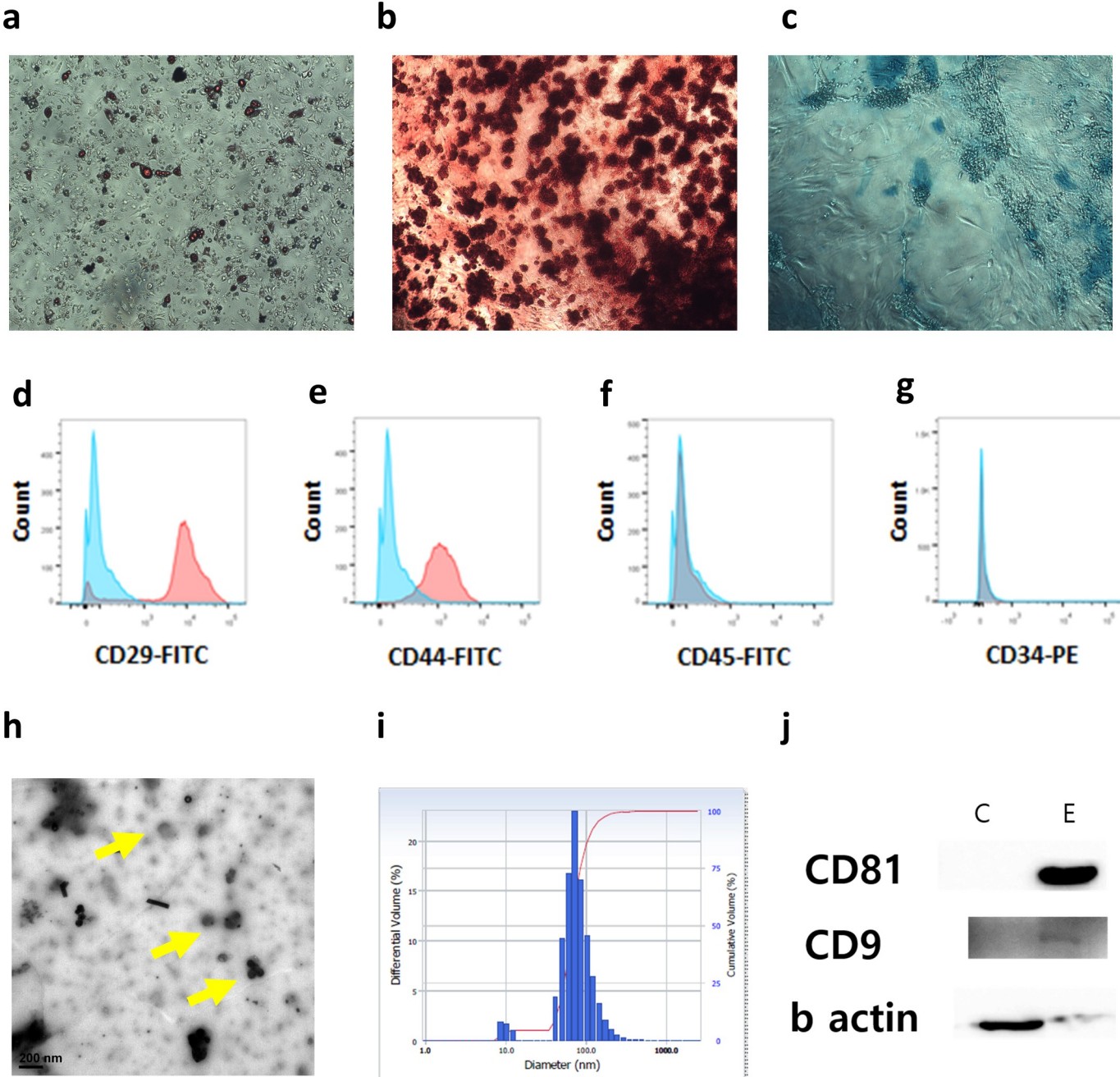

**Fig 1. cAT-MSC characterization and cAT-MSC derived EV characterization.** To identify multilineage differentiation, cAT-MSC were differentiated into **(a)** adipocyte, **(b)** osteocyte and **(c)** chondrocyte. Differentiated cells were stained each specific dye. **(d-g)** To immunotype, the markers of cAT-MCS were analyzed with CD29, CD44 as positive and CD45, CD 34 as negative. **(h)** In electron microscopic analysis, EVs were measured as 50-100nm. **(i)** Through a particles-size analyzer, the diameter of EVs was confirmed as around 100nm. **(j)** CD81 and CD 9, surface marker of EV, were confirmed as positive and β-actin was confirmed as negative in western blotting.

Using immunofluorescence of CD206, which is a marker of M2 anti-inflammatory phase, we evaluated the polarization phase of DH82 cells. Red staining was against CD11b, which is a marker of macrophage. In naïve DH82 group, red staining was visibly detected and when LPS inducing in DH82, the marker was not changed (Fig 4C and 4D). In EVs treated groups, green

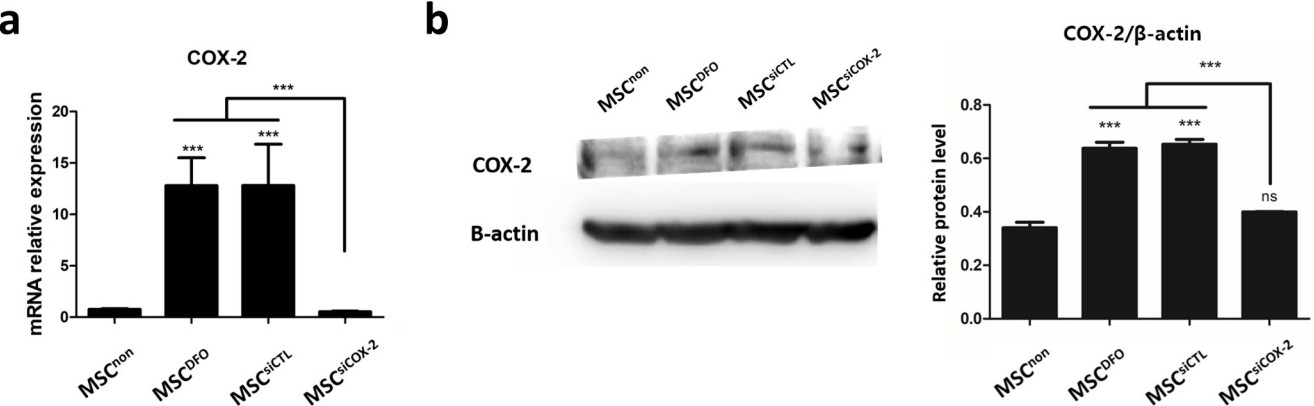

**Fig 2. The expression levels of COX-2 in cAT-MSC$^{DFO}$. (a)** The mRNA level of COX-2 is increased in DFO preconditioned and treated with siCTL group. However, COX-2 is not increased in transfected with siCOX-2 group. **(b)** The protein level of COX-2 is increased in DFO preconditioned and decreased in transfected with siCOX-2 group, same as mRNA result. Results are shown as means ± standard deviation. ***$P < 0.001$. ns, not significant.

**Fig 3. Protein levels of COX-2 in cAT-MSC derived EVs and the effect of DFO preconditioned EV in DH82. (a)** COX-2 level was increased in EVs derived from cAT-MSC$^{DFO}$ (EV$^{DFO}$). Otherwise, COX-2 was decreased in EV$^{siCOX-2}$. The result of COX-2 protein level in EVs was similar with cytosol COV-2 protein level in cAT-MSC. **(b)** Protein levels of STAT3 and p-STAT3 in DH82 treated with EVs was measured by western blotting to evaluate the effect of DFO preconditioned EVs. In the all EV treated groups, p-STAT3 (Tyr 705) was increased. Compared to EV$^{non}$ treated group, p-STAT3 is much more increased in EV$^{DFO}$ treated group, but similar in EV$^{siCOX-2}$ treated group. Results are shown as means ± standard deviation. *$P < 0.05$, ***$P < 0.001$. ns, not significant.

stain of CD206 was significantly increased and the red stain was difficult to recognize (Fig 4E–4I). Especially, CD206 was significantly increased in the groups treated with EV$^{DFO}$ (Fig 4F). The group with EV$^{siCOX-2}$ showed slight enhancement of CD206, which was not significant compared to that in the group with EV$^{DFO}$ (Fig 4H and 4I).

## Discussion

Mesenchymal stem cells (MSCs) have been studied for their anti-inflammatory therapeutic functions due to paracrine effect [19,20]. Since they interact with immune cells, including macrophages and T cells, MSC secretome is considered to be immunomodulatory and can regulate the anti-inflammatory phase [21,22]. We have recently demonstrated that paracrine effect of cAT-MSCs can significantly reprogram macrophages from M1 pro-inflammatory phase to M2 anti-inflammatory phase [18]. The study proved that secretory function of MSCs plays an important role in cell interaction, especially with inflammatory cells like macrophages. Recently, EVs were reported to play an important role among secretomes, owing to their ability to communicate and deliver substances [23–26]. Therefore, we focused on the EVs derived from cAT-MSCs. In the current study, we investigated whether DFO-preconditioned EVs have the potential to direct macrophages to M2 phase. To prove this hypothesis, HIF-1α/COX-2 axis was analyzed in cAT-MSC$^{DFO}$ and EV$^{DFO}$. After confirming that DFO could effectively increase COX-2 expression in EVs, Canine macrophage cells, DH82, were treated with EVs to confirm the delivery function of EVs and evaluate their effect on macrophage polarization.

First, accumulation of HIF-1α in the nucleus, which is an important indicator of hypoxia, was confirmed by western blot in cAT-MSC$^{DFO}$ (S1B Fig). Some reports had earlier suggested that COX-2 increases significantly with hypoxic culture under HIF-1α signal pathway [27] and is associated with both anti-inflammatory effect and macrophage polarization [28,29]. In cAT-MSC$^{DFO}$, both RNA and protein expression of COX-2 was increased, implying that DFO preconditioning could increase HIF-1α accumulation, and transcription of COX-2 was activated under HIF-1α pathway (Fig 2A and 2B).

Some reports have revealed that the same proteins may be detected either in MSC cytoplasm or in MSC secretomes and EVs [30–32]. In the current study as well, expression change of COX-2 protein in the cytosol of cAT-MSCs were also reflected in the EVs. EVs from DFO-preconditioned cAT-MSC were enriched with COX-2 molecules (Fig 3A).

HIF-1α and COX-2/PGE2 pathway are known to be associated with STAT3 activation [33,34], which is an important factor in macrophage M2 anti-inflammatory phase [35–37]. Previously published reports had suggested that macrophages are polarized into M2 phase when various molecules, such as miRNA, are delivered through EVs and stimulate STAT3 pathway in the macrophages [38–40]. Another study had revealed that not only miRNA, but proteins also can be transmitted into macrophage through EVs, and play their role [37].

The delivery system of EVs has been reported to play an important role in the anti-inflammatory effect of stem cells, especially in relation to macrophages [41]. Reprogramming macrophages from pro-inflammatory M1 phase to anti-inflammatory M2 phase is the most important mechanism in controlling immune homeostasis [42], and EVs regulate this polarization phase by transferring substances from stem cells to macrophages [43,44]. Therefore, we focused on the effect of COX-2 transferred by EVs to STAT3 signaling in macrophages. When the expression of pSTAT3 and STAT3 in LPS-induced DH82 were evaluated, these factors in LPS-induced DH82 treated with EV$^{DFO}$ were increased over than with EV$^{non}$ (Fig 3B).

The expression change of pSTAT3 in LPS-induced DH82 could be associated with macrophage phase. Thus, we evaluated macrophage phase of LPS-induced DH82 with EVs treating.

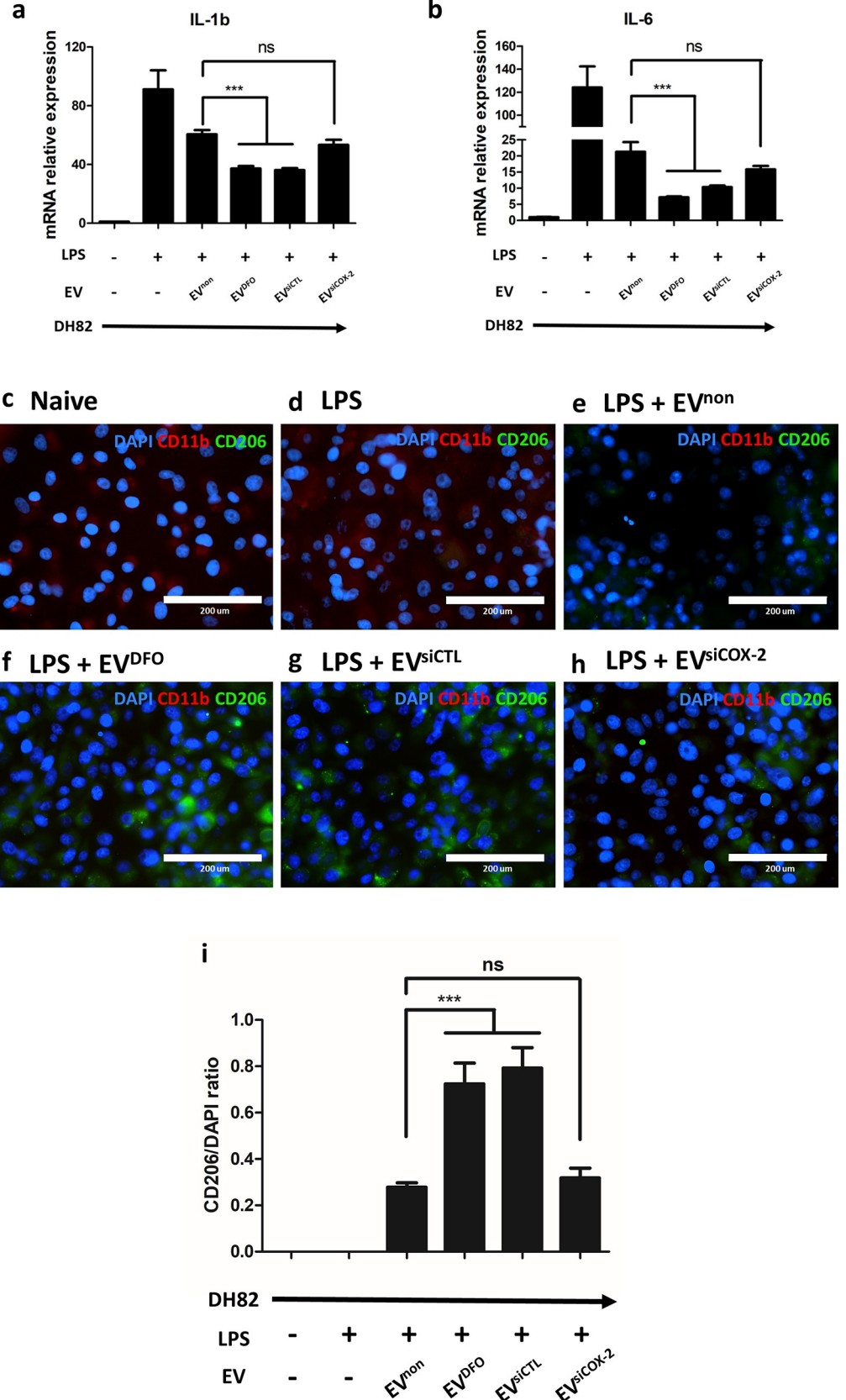

**Fig 4. The polarization phase of DH82 is directed into M2 phase when treated with EVs. (a, b)** The mRNA level of IL-1b and IL-6, which is the marker of M1 macrophage phase, was decreased in EV treated groups and significantly decreased in $EV^{DFO}$ treated group compared to $EV^{non}$ treated group. Conversely, the mRNA levels were similar in $EV^{siCOX-2}$ treated group and $EV^{non}$ treated group. **(c-h)** CD11b, which is the surface marker of macrophage, was analyzed by immunofluorescence. CD206, which is the surface marker of M2 macrophage phase, was analyzed by immunofluorescence. **(e-i)** In the EV treated groups, CD 206 with green fluorescence is increased and **(f, g)** especially more increased in $EV^{DFO}$ group. **(h)** In the $EV^{siCOX-2}$ treated group, CD206 is also increased, but not much compared to $EV^{DFO}$ group. **(i)** The ratio of CD206/DAPI. Results are shown as means ± standard deviation. $^{***}P < 0.001$. ns, not significant.

In LPS-induced DH82 cells treated with EVs, which were increased with pSTAT, the marker of M1 was reduced while that of M2 was increased (Fig 4A–4I). These changes were greater with $EV^{DFO}$. Considering all the results, EVs could deliver COX-2 to LPS-induced DH82, and might lead to M2 polarization by activating the phosphorylation of STAT3. Moreover, DFO preconditioning in cAT-MSCs enhance the macrophage directing effect through activating HIF-1α/COX-2 axis.

In this study, a limitation was that it was not clear whether substances such as miRNA in EVs could be stimulated to activate STAT3. Therefore, further studies would be required to analyze the changes in miRNA levels in cAT-MSC$^{DFO}$ and $EV^{DFO}$. However, this study found that preconditioning with DFO could affect COX-2 in cAT-MSCs and acted as anti-inflammatory molecules. $EV^{DFO}$ contained COX-2 protein and could effectively reprogram macrophage polarization into M2 phase via protein delivery system. The findings presented the therapeutic possibility of $EV^{DFO}$, which could be used in treating inflammatory diseases through macrophage reprogramming.

## Conclusions

To the best of our knowledge, this report is the first to reveal that DFO preconditioning affects EVs and macrophage polarization via EVs. DFO preconditioning increased anti-inflammatory factors, such as COX-2 in cAT-MSCs, and also increased COX-2 molecules in EVs. $EV^{DFO}$ delivered COX-2 to macrophages, which then led to M2 anti-inflammatory phase by activating STAT3 phosphorylation.

## Supporting information

**S1 Fig. (a)** Depending on DFO concentration, cell viability was not affected under 500 μM and was decreased in 1mM. **(b)** HIF-1α was accumulated in the nuclear of cAT-MSC$^{DFO}$, which showed that DFO treatment could accumulate HIF-1α in cAT-MSC nucleus. Results are shown as means ± standard deviation. $^{*}P < 0.05$. ns, not significant.
(TIF)

**S1 Raw images.**
(PDF)

## Author Contributions

**Conceptualization:** Su-Min Park.

**Investigation:** Su-Min Park.

**Resources:** Ju-Hyun An, Jeong-Hwa Lee, Kyung-Bo Kim.

**Supervision:** Woo-Jin Song, Hwa-Young Youn.

**Validation:** Hyung-Kyu Chae, Ye-In Oh, Woo-Jin Song.

**Writing – original draft:** Su-Min Park.

**Writing – review & editing:** Su-Min Park.

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
