## [Decision Letter · Decision Letter 0]

9 Mar 2021

PONE-D-21-01800

Extracellular vesicles derived from DFO-preconditioned cAT-MSCs reprogram macrophages into M2 phase

PLOS ONE

Dear Dr. Song,

Thank you for submitting your manuscript to PLOS ONE. After careful consideration, we feel that it has merit but does not fully meet PLOS ONE’s publication criteria as it currently stands. Therefore, we invite you to submit a revised version of the manuscript that addresses the points raised during the review process.

We look forward to receiving your revised manuscript.

Kind regards,

Nazmul Haque

Academic Editor

PLOS ONE

Journal Requirements:

3. We note you have included a table to which you do not refer in the text of your manuscript. Please ensure that you refer to Table 1 in your text; if accepted, production will need this reference to link the reader to the Table.

Reviewers' comments:

Reviewer's Responses to Questions

**Comments to the Author**

1. Is the manuscript technically sound, and do the data support the conclusions?

Reviewer #1: Partly

Reviewer #2: No

Reviewer #3: No

Reviewer #4: Yes

2. Has the statistical analysis been performed appropriately and rigorously? 

Reviewer #1: I Don't Know

Reviewer #2: No

Reviewer #3: No

Reviewer #4: Yes

3. Have the authors made all data underlying the findings in their manuscript fully available?

Reviewer #1: Yes

Reviewer #2: No

Reviewer #3: No

Reviewer #4: Yes

4. Is the manuscript presented in an intelligible fashion and written in standard English?

Reviewer #1: No

Reviewer #2: Yes

Reviewer #3: No

Reviewer #4: Yes

5. Review Comments to the Author

Reviewer #1: This manuscript describes the effects of DFO preconditioning on the effects of MSC-derived EVs on DH82 cells. The authors find that DFO preconditioning alters composition of EVs by increasing COX-2 levels, and STAT3 phosphorylation in target cells with concomitant M2 transition.

The manuscript has several major issue, namely

(1) Figures are all of very low resolution, which do not allow to interpret most of the results;

(2) The logic of the manuscript lack the order and should be restructured to be more reader-friendly;

(3) Experimental design and methods are poorly described;

(4) Many controls are either lacking or not mentioned:

(5) Effects of EVs on the recipient cells should be analyzed in more detailed, at least at the transcriptome level.

Other comments will be provided when high-resolution images are uploaded. Otherwise, over a half of all the results are not recognizable in the Figures. English language should be checked.

Other minor and major comments with no specific order are provided below:

Abstract:

EVnon and IF are not explained. Please, correct.

The results section in the abstract is vaguely described and does not correspond well with the conclusion. Please, make it more reader-friendly and present the most important results with the most important approaches used.

From the abstract section it is also not clear, what species of MSCs was used (human, canine?). Please, make it evident from the start.

Methods section

Lines 84-85: it is unclear when EVs were isolated and how.

Transfection of cAT-MSCs with siRNA: what is the transfection efficiency?

COX-2 knockdown should be verified by western blotting.

Isolation and characterization of EVs…: it is unclear at which point after DFO treatment EVs were isolated.

Line 173: please, change “100nm” to 100 nm

Results section

All figures have very low resolution. Please, change all images

Figure 1: with the current resolution, it is not possible to interpret this image.

Figure 2

Figure 2(a): please, change the scale to have 100% cell viability. It is the idea; you should show it does not drop significantly below 100%. How cell viability was measured? Please, describe it in detail in the Methods section.

Information about Antibodies for all targets (Lamin A, HIF-1a etc) should be provided.

The authors state that 100 uM of DFO was used for all experiments with a reference to a published manuscript. From this reasoning, experiments at Figure 2a, and b do not add anything to the manuscript. They should be removed or added to the Supplementary with according modification of the manuscript.

Figure 2c and 2d and Figure 3 and Figure 4: These images cannot be adequately interpreted due to low resolution. I will leave my comments here after the authors consider uploading images of higher resolution.

Figure 3a: COX-2 levels in EVs need to be normalized to particle number or to EVs marker. Otherwise, this data does not guarantee that the observed differences in COX-2 are related to higher COX-2 levels, not to the differences in particle numbers

Figure 3b: what are the lines corresponding to the targets? Please, clearly show it with arrows. Quantitative analysis will also be helpful, as by visual inspection the levels do not change dramatically.

Figure 4: the graphs should state the reference control for the targets

Figure 4c: the scale bar is not visible. What is the red fluorescence at all images? Red color is clearly

Figure 4c: the authors should perform quantitative analysis of CD206 staining to understand the differences between the groups. Single images are not sufficient to draw meaningful conclusions. For instance, difference between LPS+EVsictl and LPS+EVsiCOX-2 is not readily visible and may just represent arbitrary differences between visual fields.

IF analysis for CD206 measurement is also not the best option. It is necessary to perform FACS analysis and provide information about populations with CD206 intensities.

Supplementary Figure is of low resolution. It is also not structured and is hard to interpret.

Experimental schemes for every experiment is required. From the methods and results section, it is not clear how they were performed, what was the duration of the treatments etc. Please, provide such information.

What was the number of EVs for every experiment? Was it quantified and similar particle numbers added to equal amounts of DH82 cells?

Broader analysis of the DH82 transcriptome with STAT3 targets will be valuable to see the whole picture of EVs effects (altered EVs->molecular changes->functional changes)

In the Discussion section, references to the Figures are necessary.

Reviewer #2: The study by Dr Park and colleagues showed that DFO preconditioning promoted accumulation of HIF-1a in the nuclei and increased expression of COX-2 in both MSCs and MSC-EVs. The authors further suggested that EVs-DFO were able to reprogram M1 macrophages to M2 anti-inflammatory cells. The findings sound interesting, however, several concerns are raised.

Major

1. COX-2 is known as an enzyme involved in various inflammatory conditions, in particular, HIF-1a/COX-2 and its product PG are potent proinflammatory factors. However, the authors claim DFO preconditioned EVs that contain higher COX2 are anti-inflammatory. The authors need to discuss this obvious conflict.

2. LPS used in this study was reported to upregulate expression of COX-2 previously, how was COX-2 expression affected by LPS or LPS+EVs in Figs 3 and 4?

3. In inflammatory conditions, cox-2 inhibitors suppress expression of proinflammatory factors including IL-1b and IL6, however, in the current study, the authors show that DFO EVs abundant in cox-2 protein could reduce expression of these 2 proinflammatory markers in canine macrophages. How to explain these results?

4. The statistical significance was stated in Figures 2 and 4, however, there was no description of biological duplications. How many independent experiments were performed? And the authors claimed “significance” on protein expression based on WB data, however, no quantitative data were presented in figures 2 and 3.

Minor points

5. What is “IF”?

6. In figure 4, what is the red staining?

7. MSC EVs produced upon DFO precondition were reported in Ref 6, the findings should be discussed.

Reviewer #3: Park et al., isolated canine adipose tissue derived-mesenchymal stem cells (MSCs) and preconditioned with deferoxamine (DFO) in an attempt to promote their anti-inflammatory effect on macrophage M2 polarization via extracellular vesicles, the authors used different approaches to prove their hypothesis. Although the study is interesting; I have major concerns as follows:

Major concerns:

1. In Materials and methods section, isolation of MSCs was not well-described. Please describe briefly the isolation method, as MSCs are the core cells of the study.

2. Suppl figure 1 which represents the successful isolation of MSCs should be moved to Figure 1 not in the supplement.

3. The authors used antibodies against CD29, CD44, CD45, and CD34; are those adequate markers for MSCs? What about CD90, CD105, and CD73 MSC markers? There is no information about the antibodies used, are against human or canine? Is there amino acid sequence homology from human and canine for those markers to use these antibodies? Please discuss.

4. The authors used the human COX2 siRNA or control siRNA (sc-29279 and sc-37007, respectively; Santa Cruz Biotechnology, Santa Cruz, CA, USA) against canine-COX-2. Please explain.

5. Exosomal marker CD63 should be done to confirm EVs isolation. According to http://exocarta.org/Archive/ExoCarta_top100_protein_details_5.txt, ACTB is a marker for exosomes (faint band appears on the blot). The authors need non exosomal-associated proteins like calnexin or GM-130 to be done.

6. In figure 2b, there is no indication what 100, 500 numbers? Should µM and DFO added on panel

7. In figure 2 legend, '' **P < 0.01, ***P < 0.001'' is written, however, significance only *, *** as indicated on panels figure 2a&C.

8. In figure 4c-h, there is red staining, correspond to which staining?

9. To prove polarization of macrophage M1, the authors need to quantitatively quantify at least M1 markers HLA-DR and to prove reprogramming, at least M2 marker CD206 should be done by flow cytometry.

10. An experiment is needed to prove EVs internalization to mediate macrophage polarization.

Overall, the research design is not appropriate, and poor English-written manuscript. Therefore, the manuscript is not suitable to get published in Plos one Journal.

Reviewer #4: the manuscript"Extracellular vesicles derived from DFO-preconditioned cAT-MSCs reprogram macrophages into M2 phase" is well written. The experimental data are clear and specific. The discussion of the results appears well discussed and reinforced by the current literature. In my opinion the manuscript can be published without revision.

6. PLOS authors have the option to publish the peer review history of their article (what does this mean?). If published, this will include your full peer review and any attached files.

Reviewer #1: No

Reviewer #2: No

Reviewer #3: No

Reviewer #4: No

---

## [Author Response · Author response to Decision Letter 0]

3 Jun 2021

Dear Dr. Nazmul Haque

We are very pleased to have been given the opportunity to revise our manuscript entitled “Extracellular vesicles derived from DFO-preconditioned cAT-MSCs reprogram macrophages into M2 phase” for Plos one. We want to extend our appreciation to you and the reviewers for taking the time and effort necessary to provide such insightful guidance. We have carefully considered comments offered by the reviewers. Herein, we explain how we revised the research based on those comments and recommendations. The manuscript has certainly improved from these revision suggestions. We look forward to working further with you and the reviewers to move this manuscript closer to publication.

 

Reviewer #1:

1. COMMENTS: Figures are all of very low resolution, which do not allow to interpret most of the results;

RESPONSE: Thank you for raising important point. We corrected all the figures in high resolution. 

2. COMMENTS: The logic of the manuscript lack the order and should be restructured to be more reader-friendly; Experimental design and methods are poorly described; Many controls are either lacking or not mentioned:

RESPONSE: Thank you for raising important point. We corrected the manuscript to legibly and more easily understand, especially abstract and discussion. We wish this will helps to readers understand. 

3. COMMENTS: Effects of EVs on the recipient cells should be analyzed in more detailed, at least at the transcriptome level.

RESPONSE: Thank you for raising the important point. In this regard, we also think that point is a limitation of our research. However, it is a first report that COX-2 in EVs could modulate p-STAT activation in macrophage and this is involved in reprogramming of M2 phase. The findings also have significant result and based on these results, we will identify transcriptome levels such as mRNA in further study.

4. COMMENTS: [Abstract section] EV non and IF are not explained. Please, correct.

RESPONSE: Thank you for your detailed comment. We described about EVnon in page 2, lane 28. Also, we changed “IF” into “immunofluorescence” in abstracts, page 2 lane 36. 

5. COMMENTS: [Abstract section] The results section in the abstract is vaguely described and does not correspond well with the conclusion. Please, make it more reader-friendly and present the most important results with the most important approaches used.

RESPONSE: Thank you for your detailed comment. We revised the result section in abstract more legibly. We hope it helped you to understand. 

6. COMMENTS: [Abstract section] From the abstract section it is also not clear, what species of MSCs was used (human, canine?). Please, make it evident from the start.

RESPONSE: Thank you for your detailed comment. We added the “canine” in front of MSC in page 2, lane 27 & 28. We also modified the part of title as “canine AT-MSCs” to improve understanding of readers.

7. COMMENTS: [Methods section] Lines 84-85: it is unclear when EVs were isolated and how.

RESPONSE: Thank you for your detailed comment. 100 �M DFO was treated for 48 hours when the confluency of cAT-MSC was at 70-80%. After the 48 hours, the medium was collected and EVs were isolated as mentioned in page 6 lane 112-117. We corrected the sentence as “When cAT-MSCs were approximately 70% confluent, 100 �M DFO was added for 48 h in DMEM with 10% exosome-depleted FBS (Thermo Fischer Scientific, Massachusetts, USA) and 1% PS (PAN Biotech).” in page 5 lane 97-100. 

8. COMMENTS: [Methods section] Transfection of cAT-MSCs with siRNA: what is the transfection efficiency? COX-2 knockdown should be verified by western blotting.

RESPONSE: Thank you for detailed comment. In figure 2 c and d, COX-2 knock down in cAT-MSC was confirmed and we added graph that quantitively analyzed figure 2d. In siCOX-2 treating group, the mRNA and protein expression was significantly decreased, so the efficacy was proven. 

9. COMMENTS: [Methods section] Isolation and characterization of EVs…: it is unclear at which point after DFO treatment EVs were isolated.

RESPONSE: Thank you for your detailed comment. As we answered in comment 9, we corrected the sentence and this will help to understand when the EVs were isolated after DFO treatment. 

10. COMMENTS: [Methods section] Line 173: please, change “100nm” to 100 nm

RESPONSE: Thank you for your detailed comment. We changed to “100 mm” 

11. COMMENTS: [Results section] All figures have very low resolution. Please, change all images. Figure 1: with the current resolution, it is not possible to interpret this image.

RESPONSE: Thank you for raising important point. We corrected all the figures in high resolution.

12. COMMENTS: [Results section] Figure 2(a): please, change the scale to have 100% cell viability. It is the idea; you should show it does not drop significantly below 100%. How cell viability was measured? Please, describe it in detail in the Methods section. 

 The authors state that 100 uM of DFO was used for all experiments with a reference to a published manuscript. From this reasoning, experiments at Figure 2a, and b do not add anything to the manuscript. They should be removed or added to the Supplementary with according modification of the manuscript.

RESPONSE: Thank you for your detailed comment. As we accepting your comment, Figure 2 a & b were moved into supplementary figure 2. We change the Figure 2a as the scale to have 100% and there was no significant difference in cell viability between control, the groups treated with 10, 100 and 500 �M DFO. We also added the method of cell viability analysis in supplementary information. 

13. COMMENTS: Information about Antibodies for all targets (Lamin A, HIF-1a etc) should be provided.

RESPONSE: Thank you for your detailed comment. We missed the information about Lamin A antibody. Including Lamin A, all the antibodies used in this experiment were mentioned in page 7, lane 151-154. 

14. COMMENTS: Figure 2c and 2d and Figure 3 and Figure 4: These images cannot be adequately interpreted due to low resolution. I will leave my comments here after the authors consider uploading images of higher resolution.

RESPONSE: Thank you for raising important point. We corrected all the figures in high resolution. We wish that this will helps you to understand and we would appreciate if you could give an additional review. Also, Figure 2, 3 & 4 were corrected in the process of accepting other reviewer’s comment. 

15. COMMENTS: Figure 3a: COX-2 levels in EVs need to be normalized to particle number or to EVs marker. Otherwise, this data does not guarantee that the observed differences in COX-2 are related to higher COX-2 levels, not to the differences in particle numbers

RESPONSE: Thank you for raising important point. We quantified EVs and 25 μg of EVs were loaded on blot. Therefore, there was more COX-2 protein in EVDFO compared to same amount of EVnon.

16. COMMENTS: Figure 3b: what are the lines corresponding to the targets? Please, clearly show it with arrows. Quantitative analysis will also be helpful, as by visual inspection the levels do not change dramatically.

RESPONSE: Thank you for your detailed comment. The arrow means that LPS and EVs were treated in DH82 and the protein expression results have been analyzed in DH82. We also added the graph that quantified western blot result. 

17. COMMENTS: Figure 4: the graphs should state the reference control for the targets. 

Figure 4c: the scale bar is not visible. What is the red fluorescence at all images? Red color is clearly

Figure 4c: the authors should perform quantitative analysis of CD206 staining to understand the differences between the groups. Single images are not sufficient to draw meaningful conclusions. For instance, difference between LPS+EVsiCTL and LPS+EVsiCOX-2 is not readily visible and may just represent arbitrary differences between visual fields.

IF analysis for CD206 measurement is also not the best option. It is necessary to perform FACS analysis and provide information about populations with CD206 intensities.

RESPONSE: Thank you for raising the important point. We fixed the scale bar as visible. Red staining was against CD11b which is the marker of macrophage. Red staining was detected in DH82 cell lines and inducing with LPS did not changed CD11b expression. In EV treated groups, the red stain was not significantly detected and difficult to recognize, because of green fluorescence. However, CD206 alone could proves that the macrophages were not spoiled or changed into other cells. We added this description in revised manuscript in page 12, lane 247-249. 

 In this report, we focused on that EVs could induce macrophage into M2 phase. Although it was analyzed with the immunofluorescence, as we mentioned in manuscript, immunoreactive cells were calculated with 20 random fields per group as per the ratio of DAPI/CD206-positive cells to evaluate objectively. We also added figure 4i which was calculated and analyzed the ratio of DAPI/CD206-positive. 

18. COMMENTS: Supplementary Figure is of low resolution. It is also not structured and is hard to interpret.

Experimental schemes for every experiment is required. From the methods and results section, it is not clear how they were performed, what was the duration of the treatments etc. Please, provide such information.

RESPONSE: Thank you for raising important point. We corrected supplementary figures in high resolution. We also added the figure and figure legend in material and methods that explain about DFO treatment, siRNA treatment and EVs isolation. 

19. COMMENTS: What was the number of EVs for every experiment? Was it quantified and similar particle numbers added to equal amounts of DH82 cells?

RESPONSE: Thank you for raising important point. As we mentioned in page 8, lane 163, we quantified amount of EVs and DH82 cells were treated with EVs 50 μg/well for 48 h. 

20. COMMENTS: Broader analysis of the DH82 transcriptome with STAT3 targets will be valuable to see the whole picture of EVs effects (altered EVs->molecular changes->functional changes)

RESPONSE: Thank you for raising important point. We also think that broader analysis of EVs transcriptome and molecular changes in DH82 is needed. In this experiment, the possibility of DFO was shown that could improve COX-2 in EVs and these molecular changes could be transported to macrophage. It suggests that further research is very meaningful and we will identify more detailed molecules in further study. 

21. COMMENTS: In the Discussion section, references to the Figures are necessary.

RESPONSE: Thank you for your detailed comment. We added the references to the Figures in discussion section. 

 

Reviewer #2: 

1. COMMENT: COX-2 is known as an enzyme involved in various inflammatory conditions, in particular, HIF-1a/COX-2 and its product PG are potent proinflammatory factors. However, the authors claim DFO preconditioned EVs that contain higher COX2 are anti-inflammatory. The authors need to discuss this obvious conflict.

RESPONSE: Thank you for raising the important point. COX-2 is previously known as pro-inflammatory cytokine. COX-2 alone may be an inflammatory substance, but a secondary effect through macrophage is an anti-inflammatory pathway. The correlation between COX-2 and macrophage was recently revealed that COX-2 change the macrophage phase into M2 phase. These M2 macrophages activate helper T cells and secrete anti-inflammatory cytokines (Fernando O. M. et al., 2014). Thus, DFO preconditioning to canine MSC increase COX-2 expression and this cytokine could induce macrophage into M2 phase which is anti-inflammatory state. 

2. COMMENT: LPS used in this study was reported to upregulate expression of COX-2 previously, how was COX-2 expression affected by LPS or LPS+EVs in Figs 3 and 4?

RESPONSE: Thank you for detailed comment. Firstly, COX-2 was detected and analyzed in cAT-MSC and EVs. EVs that contained COX-2 was treated in LPS conditioned macrophages. In macrophage, COX-2 was not measured, but the markers of macrophage phase were analyzed such as IL-1b and IL-6. 

 LPS also induce elevation of COX-2 and STAT3 expression in macrophage (A. Baldwin Jr. et al., 1996; Xeufang Liu et al., 2018). However, considering that response of STAT3 is different between those treated only with LPS and those treated with EVs, we thought that COX-2 induced by LPS and COX-2 transferred by EVs might behave through different pathway. We are also interested in this difference and are considering further research. 

3. COMMENT: In inflammatory conditions, cox-2 inhibitors suppress expression of proinflammatory factors including IL-1b and IL6, however, in the current study, the authors show that DFO EVs abundant in cox-2 protein could reduce expression of these 2 proinflammatory markers in canine macrophages. How to explain these results?

RESPONSE: Thank you for raising the important point. COX-2 expressed in immune cells is known as initiating the inflammatory response by production of proinflammatory prostaglandins and proinflammatory cytokines. Because of this, a therapeutic strategy for inflammatory diseases has involved inhibition of COX-2, such as Non-steroidal anti-inflammatory drugs. However, recently, several reports found that COX-2–derived oxidative metabolites in activated macrophages possess anti-inflammatory (Groeger A.G. et al., 2010). COX-2 derived from MSC promote macrophage into M2 anti-inflammatory phase (Vineet K. M. et al., 2020). Thus, COX-2 derived from MSC transfer through EVs and, in this process, several substances such as membrane phospholipid may work together to reprogram macrophage into M2, which has an anti-inflammatory effect. Further research is needed on which substances are involved in M2 reprogramming.

4. COMMENT: The statistical significance was stated in Figures 2 and 4, however, there was no description of biological duplications. How many independent experiments were performed? And the authors claimed “significance” on protein expression based on WB data, however, no quantitative data were presented in figures 2 and 3.

RESPONSE: Thank you for raising the important point. All experiment was performed three time as duplicate. Each quantitative data is added on Figure 2 and 3. 

5. COMMENT: What is “IF”?

RESPONSE: Thank you for detailed comment. We changed “IF” into “immunofluorescence” in abstracts, page 2 lane 36. 

6. COMMENT: In figure 4, what is the red staining?

RESPONSE: Thank you for detailed comment. Red staining was against CD11b which is the marker of macrophage. Red staining was detected in DH82 cell lines and inducing with LPS did not changed CD11b expression. In EV treated groups, the red stain was not significantly detected and difficult to recognize, because of green fluorescence. However, CD206 alone could proves that the macrophages were not spoiled or changed into other cells. We added this description in revised manuscript in page 12, lane 247-249. 

7. COMMENT: MSC EVs produced upon DFO precondition were reported in Ref 6, the findings should be discussed.

RESPONSE: Thank you for detailed comment. We changed the sentence as “Several studies have shown deferoxamine (DFO), a hypoxia mimetic agent, to be usable in hypoxia preconditioning [7] and improve angiogenesis effect of MSC-derived EVs [8]”. Also the reference number was changed as Ref 8. 

 

Reviewer #3 

1. COMMENT: In Materials and methods section, isolation of MSCs was not well-described. Please describe briefly the isolation method, as MSCs are the core cells of the study.

RESPONSE: Thank you for your detailed comment. We added isolation method of canine adipose derived MSC briefly in page 4 lane 78. 

2. COMMENT: Suppl figure 1 which represents the successful isolation of MSCs should be moved to Figure 1 not in the supplement.

RESPONSE: Thank you for your kind comment. Accepting your comment, supplement figures were moved to figure 1 and also the relevant content in manuscript has been revised in page 9, lane 177-182. 

3. COMMENT: The authors used antibodies against CD29, CD44, CD45, and CD34; are those adequate markers for MSCs? What about CD90, CD105, and CD73 MSC markers? There is no information about the antibodies used, are against human or canine? Is there amino acid sequence homology from human and canine for those markers to use these antibodies? Please discuss.

RESPONSE: Thank you for your detailed comment. The markers of canine mesenchymal stem cell were revealed as positive in CD 29, CD 44, CD 73, CD 90 and negative in CD 34, CD 45 and MCH-II (Ana I. et al., 2017; Keith A. R. et al., 2016). CD 73 also showed moderate positive expression in canine MSC, but not strong. The antibodies used in this experiment were all for canine specific antibodies, and the antibodies available for purchase were selected and experimented. Specific antibody information is described in supporting information. 

4. COMMENT: The authors used the human COX2 siRNA or control siRNA (sc-29279 and sc-37007, respectively; Santa Cruz Biotechnology, Santa Cruz, CA, USA) against canine-COX-2. Please explain.

RESPONSE: Thank you for your detailed comment. Since there is no commercial siRNA available for canine. However, it was found to be sufficiently useful considering that in present report, mRNA and protein expression of COX-2 was significantly decreased using COX-2 siRNA. 

5. COMMENT: Exosomal marker CD63 should be done to confirm EVs isolation. According to http://exocarta.org/Archive/ExoCarta_top100_protein_details_5.txt, ACTB is a marker for exosomes (faint band appears on the blot). The authors need non exosomal-associated proteins like calnexin or GM-130 to be done.

RESPONSE: Thank you for detailed comment. As far as we know that currently there is no consensus about general markers of EVs. The several markers were confirmed and in several papers, β-actin is used as negative marker of exosome (Shin L. S. et al., 2020; Lin C. et al., 2020; Hairong W. et al., 2018). Also, we used CD81 and CD9 as positive marker of EVs and these makers are broadly used as typical markers of EVs. Therefore, these markers have already been certified in other papers, and we have also verified by measuring size of EVs. So we determined that EVs, which were separated, have no problem with proceeding experiment. We hope this will help to persuade you. 

6. COMMENT: In figure 2b, there is no indication what 100, 500 numbers? Should µM and DFO added on panel

RESPONSE: Thank you for detailed comment. We moved the figure 2b to supplementary figure 2 by accepting comments from the other reviewer. Figure was corrected as marking the concentration of DFO. 

7. COMMENT: In figure 2 legend, '' **P < 0.01, ***P < 0.001'' is written, however, significance only *, *** as indicated on panels figure 2a&C.

RESPONSE: Thank you for detailed comment. We moved Figure 2a &b to supplementary figure 1 as accepting the comment of the other reviewer. Thus, we deleted “**P < 0.01” in figure 2 legend and in supplementary figure legend, fixed as *P < 0.05. 

8. COMMENT: In figure 4c-h, there is red staining, correspond to which staining?

RESPONSE: Thank you for detailed comment. We fixed the scale bar as visible. Red staining was against CD11b which is the marker of macrophage. Red staining was detected in DH82 cell lines and inducing with LPS did not changed CD11b expression. In EV treated groups, the red stain was not significantly detected and difficult to recognize, because of green fluorescence. However, CD206 alone could proves that the macrophages were not spoiled or changed into other cells. We added this description in revised manuscript in page 12, lane 247-249. 

9. COMMENT: To prove polarization of macrophage M1, the authors need to quantitatively quantify at least M1 markers HLA-DR and to prove reprogramming, at least M2 marker CD206 should be done by flow cytometry.

RESPONSE: Thank you for raising the important point and thank you for letting me know the important approaches. In this report, we focused on that EVs could induce macrophage into M2 phase. Although it was analyzed with the immunofluorescence, as we mentioned in manuscript, immunoreactive cells were calculated with 20 random fields per group as per the ratio of DAPI/CD206-positive cells to evaluate objectively. We also added figure 4i which was calculated and analyzed the ratio of DAPI/CD206-positive.

10. COMMENT: An experiment is needed to prove EVs internalization to mediate macrophage polarization.

RESPONSE: Thank you for raising the important point. We also thought about internalization, and in this experiment, we thought that the internalization effect of EVs to macrophage could be indirectly inferred through the difference of macrophage phase and p-STAT3 between groups with and without COX-2 in the EVs. 

Reviewer #4: the manuscript "Extracellular vesicles derived from DFO-preconditioned cAT-MSCs reprogram macrophages into M2 phase" is well written. The experimental data are clear and specific. The discussion of the results appears well discussed and reinforced by the current literature. In my opinion the manuscript can be published without revision.

COMMENT: Thank you for your agreement.

Sincerely yours,

Woo-Jin, Song

Professor, DVM, Ph.D

Laboratory of veterinary internal medicine, College of veterinary medicine,

Jeju National University 

Jeju 63243, Republic of Korea

and

Hwa-Young Youn

Professor, DVM, Ph.D

Laboratory of veterinary internal medicine, College of veterinary medicine,

Seoul National University

Seoul 08826, Republic of Korea

---

## [Decision Letter · Decision Letter 1]

1 Jul 2021

Extracellular vesicles derived from DFO-preconditioned cAT-MSCs reprogram macrophages into M2 phase

PONE-D-21-01800R1

Dear Dr. Song,

We’re pleased to inform you that your manuscript has been judged scientifically suitable for publication and will be formally accepted for publication once it meets all outstanding technical requirements.

Kind regards,

Nazmul Haque

Academic Editor

PLOS ONE

Additional Editor Comments (optional):

Reviewers' comments:

Reviewer's Responses to Questions

**Comments to the Author**

1. If the authors have adequately addressed your comments raised in a previous round of review and you feel that this manuscript is now acceptable for publication, you may indicate that here to bypass the “Comments to the Author” section, enter your conflict of interest statement in the “Confidential to Editor” section, and submit your "Accept" recommendation.

Reviewer #1: All comments have been addressed

Reviewer #3: All comments have been addressed

2. Is the manuscript technically sound, and do the data support the conclusions?

Reviewer #1: Yes

Reviewer #3: Yes

3. Has the statistical analysis been performed appropriately and rigorously? 

Reviewer #1: Yes

Reviewer #3: Yes

4. Have the authors made all data underlying the findings in their manuscript fully available?

Reviewer #1: Yes

Reviewer #3: (No Response)

5. Is the manuscript presented in an intelligible fashion and written in standard English?

Reviewer #1: Yes

Reviewer #3: Yes

6. Review Comments to the Author

Reviewer #1: The authors addressed the major concerns and erased points. In its current form, the manuscript can be accepted for publication.

Reviewer #3: All comments have been addressed. No further concerns do exist. The manuscript can be accepted in its present form.

7. PLOS authors have the option to publish the peer review history of their article (what does this mean?). If published, this will include your full peer review and any attached files.

Reviewer #1: No

Reviewer #3: No

---

## [Editor Report · Acceptance letter]

15 Jul 2021

PONE-D-21-01800R1 

Extracellular vesicles derived from DFO-preconditioned canine AT-MSCs reprogram macrophages into M2 phase 

Dear Dr. Song:

I'm pleased to inform you that your manuscript has been deemed suitable for publication in PLOS ONE. Congratulations! Your manuscript is now with our production department. 

Kind regards, 

on behalf of

Dr. Nazmul Haque 

Academic Editor

PLOS ONE